# The Application of a Sodium Benzoate Salt-Nucleating Agent in Recycled Polyethylene Terephthalate: Crystallization Behavior and Mechanism

**DOI:** 10.3390/molecules30010037

**Published:** 2024-12-26

**Authors:** Meizhen Wang, Fuhua Lin, Tianjiao Zhao, Yapeng Dong, Xinyu Hao, Dingyi Ning, Yanli Zhang, Kexin Zhang, Dan Zhou, Jun Luo, Xiangyang Li, Bo Wang

**Affiliations:** 1School of Chemical Engineering and Technology, Taiyuan University of Science and Technology, Taiyuan 030024, China; s202321111073@stu.tyust.edu.cn (M.W.); s202221211044@stu.tyust.edu.cn (T.Z.); s202221211035@stu.tyust.edu.cn (Y.D.); s202421211097@stu.tyust.edu.cn (X.H.); s202421211098@stu.tyust.edu.cn (D.N.); 2021051@tyust.edu.cn (Y.Z.); 2022072@tyust.edu.cn (K.Z.); 2021053@tyust.edu.cn (D.Z.); 2School of Traffic Engineering, Shanxi Vocational University of Engineering Science and Technology, Jinzhong 030619, China; linfuhua@sxgkd.edu.cn; 3Guangzhou Fibre Product Testing and Research Institute, Guangzhou 510220, China; luoj@iccas.ac.cn; 4Department of Chemical and Chemical Engineering, Shanxi Polytechnic College, Taiyuan 030006, China

**Keywords:** recycled polyethylene terephthalate, chemical nucleation, crystallization behavior

## Abstract

The molecular chains of recycled polyethylene terephthalate (rPET) show breakage during daily use, causing poor crystallization and leading to mechanical properties that, when blended with the nucleating agent, become an effective method of solving this problem. The salt-nucleating agent sodium benzoate (SB), disodium terephthalate (DT), and trisodium 1,3,5benzene tricarboxylic (TBT) were synthesized, and an rPET/nucleating agent blend was prepared. The intrinsic viscosity (*η*) results showed that the *η* of the rPET/SB was decreased, which indicated the breakage of the rPET molecular chains. The FTIR results indicated that a chemical reaction occurred between the rPET and Na^+^ of the SB. Moreover, the Na^+^ content of the DT and TBT were higher than that of the SB, which increased the opportunity for low-molecular-weight rPET to reattach to the organic carboxylic acid portion of the nucleating agent, thereby increasing the *η* of the rPET/DT and rPET/TBT. The salt-nucleating agent sodium benzoate greatly improved the crystallization properties of the rPET, resulting in the half-crystallization time decreasing, the crystallization temperature increasing, and the effect of SB being better than that of DT and TBT. This was because the nucleating agent caused chemical nucleation with rPET, and the ionic groups acted as nucleation sites, while the rPET/DT and rPET/TBT, which had high molecular weights, hindered the improvement of the crystallization properties. The mechanical properties prove that the rPET/SB decreased due to the severe degradation of the rPET molecular chains. The mechanical properties of the rPET/DT and rPET/TBT were effectively improved because of the nucleating agent refining the grain size of the rPET and the high molecular weight. But the stacking of multitudinous rPET molecular chains can form a structure resembling physical cross-linking, causing a slight decrease in the mechanical properties of the rPET/TBT compared to the rPET/DT.

## 1. Introduction

Polyethylene terephthalate (PET) has the advantages of favorable heat and corrosion resistance properties and is therefore widely used in food packaging, plastic beverage bottles, synthetic fibers, and other fields [1,2]. Currently, the global use of PET is enormous, with the consumption of plastic beverage bottles having increased by 100 billion units from 2016 to 2021 [3]. Plastic beverage bottles are difficult to be degrade using microorganisms in a short period of time in their natural state, while their random disposal after use will cause serious white pollution [4,5]. In order to alleviate the environmental pollution caused by PET, physical and chemical recycling methods have been applied to the recycling and treatment of waste PET [6]. Chemical recycling involves the use of ester bond cleavage to depolymerize PET into oligomeric or monomeric states, or to prepare products with higher added value [7]. Physical recycling involves processes such as crushing, cleaning, drying, granulation, and reprocessing the waste PET to produce recycled PET (rPET). Compared with chemical recycling, physical recycling has the advantages of a simple process and the low cost of processing equipment. Therefore, the physical recycling of PET has become the most widely used PET recycling method, and rPET, as a secondary raw material, has been widely studied.

rPET is a structured linear polymer with the same molecular chain symmetry as PET. The repeating units of rPET molecular chains are primarily methylene and a rigid conjugated system, which comprise benzene rings and ester groups [8,9]. This distinctive conjugation system, while imparting rigidity to rPET, hinders the motion activity of rPET molecular chain segments during the crystallization process, leading to a diminished crystallization rate [10,11]. Although methylene can bring flexibility to rPET, the methylene structure is too short to change the effect of the rigid conjugated system on the crystallization of rPET [12]. Most importantly, the thermal degradation and photolysis of PET during daily use can lead to a decrease in its molecular weight and mechanical properties [13]. Liu et al. [14] utilized the chain extension effect of polyphenylmethyl isocyanate (PMDI) to modify PET. The results showed that the addition of PMDI effectively increased the molecular weight of PET, thereby improving the melt strength and tensile strength due to the increase in molecular chain entanglement and interactions. Meanwhile, some studies have shown that a lower molecular weight increases the fluidity of rPET molecular chains, which facilitates their rearrangement and the formation of crystalline structures, thereby enhancing the crystallinity of rPET [15]. Therefore, seeking effective methods to enable rPET to possess both good crystallization and mechanical properties are key to expanding the application of rPET [16]. At present, there are fewer studies on the modification of rPET by the addition of a nucleating agent, which must be adapted from PET modification studies. The modification methods of PET include copolymerization, blending, and various synergistic effects. Among them, blending modification is widely used in the field of PET modification due to its advantages of simple operation and high cost-effectiveness [17].

It was found that sodium benzoate (SB) (Figure 1b) can serve as an effective nucleating agent for PET in previous research, and that its nucleation mechanism with PET was chemical nucleation (Figure 1a) [18]. During the nucleation process, the ends of the PET molecular chains react with the Na^+^ of the SB, forming an anionic end group. The anionic end group can play a good role in the nucleation process of PET, as the nucleation sites can accelerate the crystallization rate of PET [19]. Meanwhile, chemical nucleation can cause the breakage of PET molecular chains, creating PET with a small molecular weight [20]. The molecular chain mobility of PET with a small molecular weight increased, which caused the further acceleration of the crystallization rate of PET [21,22]. However, the localized decrease in molecular weight of PET led to a decrease in the mechanical and thermal properties of PET [23,24].

Sodium p−hydroxybenzoate (PHB-2Na) is an organic carboxylate salt with two Na^+^ (Figure 1c). Its nucleation mechanism with PET is similar to that of SB, as shown in Figure 1a. The research indicates that PHB-2Na is slightly inferior to SB in improving the crystallization ability of PET. But the PHB-2Na can maintain the degradation of PET at a lower level, making the mechanical properties of PET/PHB-2Na more excellent than those of PET/SB [25]. Disodium salt of bicyclol [2.2.1] heptane-2,3-dicarboxylic acid (HPN-68L) is also an organic carboxylate salt with two Na^+^ (Figure 1d). Its nucleation mechanism with PET is similar to that of SB, as shown in Figure 1a. The results indicate that the addition of HPN-68L can effectively improve the crystallization behavior of PET, but the modification effect is also inferior to that of SB. Meanwhile, the addition of HPN-68L reduces the equilibrium melting point of the blend while increasing its flexural strength [26].

From the perspective of chemical nucleation, theoretically, the more Na⁺ ions there are, the more nucleation sites will be available for chemical nucleation, which leads to a greater degree of molecular chains breakage and a better improvement effect on the crystallization behavior of PET. Nevertheless, the aforementioned studies have indicated that a larger quantity of Na⁺ ions is not necessarily conducive to enhancing the crystallization behaviors of PET. Instead, it has a certain impact on improving some of the mechanical properties of PET. Due to the differences in the organic carboxylic acid components of SB, PHB-2Na, and HPN-68L in the aforementioned study, the amounts of Na^+^ also vary. It remains unclear how these two factors influence the crystallization behavior of PET through chemical nucleation. Therefore, it becomes crucial to explore how chemical nucleation plays a role in the crystallization behavior of PET under the influence of a single variable, and to extend the chemical nucleation mechanism to the modified application field of rPET, to exploit the new rPET nucleating agent and obtain high-performance rPET, which has also become the innovation point of this research.

In this study, three kinds of sodium benzoate salt nucleating agents were synthesized by chemical reaction with the same organic carboxylic acid structure and the different Na^+^ amounts, namely, SB, disodium terephthalate (DT), and trisodium 1,3,5-benzene tricarboxylate (TBT). The three kinds of sodium benzoate salt nucleating agents were separately subject to melting blending with rPET to prepare the rPET/sodium benzoate salt nucleating agent blend. The sodium benzoate salt nucleating agent and the blend were subjected to detailed characterization using Fourier-transform infrared spectroscopy (FTIR). The thermal stability of the nucleating agent was evaluated using a thermogravimetric analyzer (TGA). The relative molecular weight of the rPET/sodium benzoate salt nucleating agent blend was characterized by using a torque rheometer and a UST viscometer. The crystallization behavior of the rPET/sodium benzoate salt nucleating agent blend was investigated by differential scanning calorimetry (DSC). In parallel, the nucleation mechanism of chemical nucleation and the effect of chemical nucleation on the crystallization behavior of rPET were studied in depth. Moreover, the mechanical properties of the blend were also discussed.

## 2. Result

### 2.1. The Characterization Analysis of the Sodium Benzoate Salt Nucleating Agent

The FTIR spectra of the sample is shown in Figure 2a. The vibration peaks of the benzene ring skeleton present at 1600, 1610, and 1617 cm^−1^ [27]. The vibration peaks of the methyl C-H bending vibration present at 1412, 1387, and 1371 cm^−1^. The vibrational peaks of COO^−^ present at 1553, 1562, and 1575 cm^−1^. The peak at 709 cm^−1^ represents a single substitution in the benzene ring, the peak at 737 cm^−1^ represents a para-substitution in the benzene ring, and the peak at 752 cm^−1^ represent a 1, 3, and 5 triple substitution in the benzene ring [28,29]. The results prove that the sodium benzoate salt nucleating agents were synthesized successfully.

The thermogravimetric curves of the sample are shown in Figure 2b. It can be observed that all the sodium benzoate salt nucleating agents demonstrate excellent thermal stability and do not undergo thermal decomposition within the temperature range of 100–300 °C. The results indicate that the chemical structure of the nucleating agent can remain intact at the processing temperature of rPET (280 °C), and thus can be used as a dispersive nucleating agent for rPET.

### 2.2. The Chemical Nucleation of the rPET/Sodium Benzoate Salt Nucleating Agent

The FTIR spectra of the sample are shown in Figure 3a. The C=O stretching vibrational peak appears at approximately 1715 cm^−1^. The O-C=O stretching vibration peak appears at 1270 cm^−1^. The backbone vibrational absorption peaks of the benzene ring appear at 1408 cm^−1^, and the out-of-plane bending vibrational absorption peak of the benzene ring C-H appears at 872 cm^−1^ [30]. The results demonstrate that the backbone structure of the benzene ring of rPET remains unaltered following the addition of the nucleating agent. However, compared with the pure rPET, rPET/SB, rPET/DT, and rPET/TBT show new unique absorption peaks COO^−^ at 1552 cm^−1^ and 1460 cm^−1^ [31,32]. The results indicate that the sodium benzoate salt nucleating agent has undergone a successful reaction with rPET.

The solubility parameters of the sample are shown in Table 1. The solubility parameter reflects the magnitude of the intermolecular forces within a substance and can be used as a parameter to quantitatively characterize the interaction between the polymer and the solvent. The closer the values of the two solubility parameters are, the more compatible the two substances will be, and the better the solvent will be able to dissolve the polymer. The basis of the chemical nucleation lies in the fact that the nucleating agent and rPET can undergo a chemical reaction. Then, the reactivity strength of the two reactants is of great significance as it materializes chemical nucleation. The solubility parameter of the nucleating agent plays a decisive role in the formation of the polymer ionic salt [33]. According to Table 1, the solubility parameter values of benzoic acid and rPET are the closest. This indicates that sodium benzoate and rPET have good compatibility, which enables chemical reactions to occur more easily and uniformly. In contrast, the solubility parameter values of terephthalic acid and 1,3,5-benzenetricarboxylic acid are larger than those of benzoic acid. Therefore, the sodium salts of these two acids and rPET have a weaker reaction ability than the SB. As a result, their nucleation ability is inferior to that of SB.

The so-called chemical nucleation refers to the reaction between the ends of rPET molecular chains and a sodium carboxylate nucleating agent to produce a new substance with an ionic end group. This substance is the real nucleation site that improves the crystallize properties of rPET, rather than the nucleating agent itself. As shown in Figure 3b, a large number of ionic end groups come together to form ionic clusters. When these ionic clusters aggregate, the electrostatic interactions between the anions and cations overcome the spatial site-barrier effect. This brings the covalently bonded chain segment to the vicinity of the spatially narrow ionic clusters and cause the chain segment to stack tightly [34,35]. In other words, each -COO^−^ group with the ionic cluster has a covalently linked rPET chain segment attached to it. If a large number of rPET molecular chains segment are clustered around the periphery of the ion cluster, then the ion cluster and the rPET molecular chains segment must adjust to a more compact state, forming an ordered structure similar to micelles [36]. Therefore, the space available for free movement between the rPET molecular chain segments is significantly reduced, strongly promoting crystal nucleation, which is the underlying cause of nucleation.

### 2.3. The Viscosity and Torque Testing of the rPET/Sodium Benzoate Salt Nucleating

The intrinsic viscosity and relative molecular weight of the rPET/sodium benzoate salt nucleating agent blend are presented in Table 2. The curve of the torque over time is illustrated in Figure 4a. The ends of the rPET molecular chains undergo chemical nucleation with SB, and the reaction mechanism is shown in Figure 4b. This reaction produces a new substance, rPET-COONa with ionic end groups, and benzoate-capped rPET, resulting in the breakage of the rPET molecular chains. This breakage leads to a decrease in the local viscosity average molecular weight (*M_η_*) and intrinsic viscosity (*η*) of rPET [37].

Compared with SB, DT and TBT possess a higher Na^+^ content. After chemical nucleation leads to the breakage of the rPET molecular chains, more low-molecular-weight rPET can attach to the organic carboxylic acid part of the nucleating agent, giving rPET a larger molecular chain structure and increasing its molecular weight. Therefore, the *η* of rPET/DT and rPET/TBT is that of pure rPET. Meanwhile, a higher molecular weight can effectively improve the mechanical properties of the rPET, which can also be confirmed in a subsequent mechanical performance test. Similarly, as shown in Figure 4a, the equilibrium torque of low-molecular-weight rPET/SB is lower than that of rPET, whereas rPET/DT and rPET/TBT, due to their higher molecular weights, exhibit a larger equilibrium torque than pure rPET.

### 2.4. Non-Isothermal Crystallization Behavior of the rPET/Sodium Benzoate Salt Nucleating Agent Blend

The non-isothermal crystallization behaviors of the rPET/sodium benzoate salt nucleating agent blend are presented in Figure 5, and the crystallization parameters of the rPET/sodium benzoate salt nucleating agent blend composite are listed in Table 3. Compared to rPET, the cold crystallization temperature (*T_c__c_*) of the rPET/sodium benzoate salt nucleating agent blend is shifted towards a lower temperature. This phenomenon can be attributed to the interaction between the surface of the nucleating agent and the molecular chains of rPET, which facilitates the orientation of the molecular chains on the surface of the nucleating agent. This enables rPET to crystallize at a lower temperature, thus accelerating the crystallization rate of rPET [38,39].

Compared to the rPET, the crystallization temperature (*T_c_*) of the rPET/sodium benzoate salt nucleating agent blend increases by 26.03 °C, 14.6 °C, and 9.85 °C, respectively. This is because the addition of the nucleating agent can reduce the free energy required for the formation of critical nuclei, promote the formation of crystal nuclei and the growth of crystals, enabling the rPET molecular chains to begin to arrange themselves and crystallize in an orderly manner at higher temperatures. The half peak width of the crystallization peak is related to the crystallization rate and grain size. Under the same conditions, the narrower the half peak width of crystallization, the faster the crystallization rate and the more uniform grain size [40]. As shown in Figure 5b, it can be observed that the half-width of the crystallization peak of the rPET/sodium benzoate salt nucleating agent blend is narrower than that of the pure rPET. Among them, the half-width of the crystallization peak of the rPET/SB is the narrowest, being approximately half that of the pure rPET, which indicates that SB exhibits the strongest nucleation ability for rPET and the resulting spherocrystal is more perfect. 

The results in Table 3 indicate that the addition of SB improved the compliance of rPET, causing a 1.65 °C reduction in the melting temperature (*T_m_*) of rPET/SB compared with that of rPET. This is because SB reacts with the ends of the rPET molecular chains, disrupting the integrity of the rPET molecular chains and causing local breakage. As a result, the interaction force between the rPET molecular segments weakens and the melting temperature decreases. The *T_m_* of the rPET/DT and rPET/TBT increased by 0.86 °C and 1.13 °C compared with that of rPET. This indicates that the reaction of rPET with DT and TBT increases the relative molecular weight and crystal thickness of the rPET molecular chains. And the thicker crystal can reduce the degree of freedom of the molecular chain segments, enhance the intermolecular interactions of the polymer, and promote the growth of the rPET crystal.

The degree of crystallinity (*X_c_*) of the sample is calculated using Equation (1) and is presented in Table 3.
(1)$$Xc=100\%\times \frac{\Delta H_{m}}{{\Delta H}_{m}^{0}}$$
where ${∆H}_{m}$ is calculated from the areas under the melting peak, and ${∆H}_{m}^{0}$ is the melting enthalpy of 100% crystallization rPET, which is 140.1 J/g [41].

According to Table 3, the crystallinity of the rPET/SB, rPET/DT, and rPET/TBT increases by 5.5%, 3.5%, and 2.5% compared to that of rPET. This is because the Na^+^ of the nucleating agent undergoes chemical reaction with the ends of the rPET molecular chains, generating a new substance, rPET-COONa, with ionic end groups. This substance serves as a nucleation site and can induce the formation of more crystal nuclei which promotes the arrangement of the rPET molecular segments in a more ordered manner, making more crystalline regions and higher crystallinity. Meanwhile, the nucleating agent induces rPET to crystallize at higher temperatures by reducing the free energy barrier of crystallization. At higher temperatures, the rPET can obtain more energy, enabling the movement of the rPET molecular segments to be more active and easier to arrange into ordered crystalline structures, thus enhancing the crystallinity.

### 2.5. Isothermal Crystallization Behavior of the rPET/Sodium Benzoate Salt Nucleating Agent Blend

Figure 6 illustrates the isothermal crystallization curve of the rPET/sodium benzoate salt nucleating agent blend at 225 °C. Figure 6b uses the Avrami equation to convert the isothermal crystallization curves of each sample into curves of relative crystallinity and crystallization time to describe the speed of the isothermal crystallization process. The Avrami equation is shown in Equation (2).
(2)$$X(t)=\exp(-K_{t}t^{n})$$
where “*X*(*t*)” is the relative crystallinity obtained by the integrals at time *t*, *K* is the crystallization rate constant, and *n* is the Avrami exponent relating to the crystal growth geometry.

When *X*(*t*) = 50%, by combining n and k, the Avrami equation can be used to obtain the calculation formula for the corresponding half crystallization time (*t*_1/2_), which is shown in Equation (3) [42]. Another form of the Avrami equation can be represented by Equation (4):(3)$$t_{1/2}={\left(\frac{\ln2}{K}\right)}^{1/n}$$
(4)ln[−ln(1−X(t)]=lnK+nlnt
where the slope of the straight line is *n* and the intercept is *K*.

According to Table 4, the *t*_1/2_ of the rPET/SB, rPET/DT, and rPET/TBT are shortened by 12.1 min, 8.54 min, and 4.71 min, respectively. Compared with the study by Zhao et al. [43], in which the addition of the nucleating agent shortened the *t*_1/2_ of rPET by 3.49 min, the addition of the nucleating agent in this study significantly accelerated the crystallization rate of rPET. This indicated that the nucleating agent reacts with the rPET molecular chains to form ionic groups with nucleating properties. The ionic groups are prone to forming thermodynamically stable crystal nuclei of a certain size at higher temperatures, promoting the rapid growth of subsequent crystal nuclei. In addition, after the reaction between the SB and rPET molecular chains breaks, the low-molecular-weight rPET segments possess stronger ability and are easier to diffuse into the lattice than the original rPET, thereby promoting the crystallization of the entire blend due to accelerated local crystallization [44]. However, the rPET/DT and rPET/TBT possess high molecular weights, which, to some extent, limits the crystallization rate of the blending. The combined effect of these two aspects significantly shortened the crystallization induction period of the rPET/SB, enabling the rPET/SB to have the fastest crystallization rate in comparison with the rPET/DT and rPET/TBT.

The intercept *K* is the crystallization rate constant, which reflects the crystallization rate of the sample. The Avrami index *n* is related to the nucleation mechanism and the growth mode, which can be expressed as the sum of the spatial dimension of growth and the time dimension of nucleation. As shown in Table 4, the Avrami index *n* of both the pure rPET and the rPET/sodium benzoate salt nucleating agent blend are approximately 3. This indicates that the nucleation and crystal growth mechanisms of the pure rPET and the rPET/sodium benzoate salt nucleating agent blend are highly similar, and the crystal growth process follows three-dimensional crystal growth. This also implies that the addition of nucleating agent did not alter the crystal growth mode of rPET.

### 2.6. The POM of the rPET/Sodium Benzoate Salt Nucleating Agent Blend

The POM morphology of the sample following isothermal crystallization at 225 °C is illustrated in Figure 7. It can be observed that the distribution of spherulites in the rPET is sparse and the sizes are diverse. The morphology of the rPET/sodium benzoate salt nucleating agent blend presents a uniformly distributed fine grain structure. This indicates that the nucleating agent can effectively improve the uniformity of the rPET grain, making an increase in the number and density of spherulites [45]. The ionic end groups are generated by the reaction between the nucleating agent and the end groups of the rPET molecular chains. When a large number of ion end group cluster, the segments of the rPET molecular chains segment are arranged more tightly and orderly, thus making the distribution of the rPET grains more uniform. At the same observation scale, the nucleating agent can refine the grain size of the rPET matrix, thereby improving the mechanical properties.

### 2.7. The Mechanical Properties of the rPET/Sodium Benzoate Salt Nucleating Agent Blend

The mechanical property data of the rPET/sodium benzoate nucleating agent blend is shown in Table 5. Compared with the rPET, the tensile strength, flexural strength, flexural modulus, and Izod notched impact strength of the rPET/SB decreases by 12.26 MPa, 25.96 MPa, 193.07 MPa, and 0.22 kJ/m^2^, respectively. The mechanical properties of the rPET/DT and rPET/TBT show an upward trend in comparison with the pure rPET. This is because of chemical nucleation between the SB and rPET, which causes severe breakage of the rPET molecular chains and disrupts the stress transmission path within rPET, preventing it from effectively resisting tensile deformation when subjected to external forces. At the same time, the breakage of the rPET molecular chains leads to a reduction in the intermolecular interaction forces, resulting in a decrease in impact toughness and rigidity. Although the addition of SB optimizes the internal molecular conformation of rPET and accelerates the relaxation and reconstruction regularity of the molecular chain, significantly improving the crystallinity of rPET, the severe breakage of the molecular chains still leads to a decrease in the flexural modulus of rPET. The addition of the nucleating agent can refine the crystal size of rPET, making an increase in crystal density [46]. By promoting the ordered arrangement of the rPET molecular chains and improving the crystal structure of rPET, the flexural modulus of the rPET/DT and rPET/TBT are significantly increased, which is consistent with the increase in crystallinity shown in Table 3. Meanwhile, the rPET-COONa formed during chemical nucleation process has a certain adsorption effect on the rPET molecular chains, which can tightly stack the rPET molecular chains segments near the ion cluster. During the process of tightly stacking rPET molecular chains, a structure resembling physical cross-linking is formed. By forming cross-linking points, the molecular chains of rPET are more effectively arranged, reducing the presence of disordered and defective regions [47]. This enables the rPET to better resist deformation and damage when subjected to external forces, thereby improving its mechanical properties. However, due to the high Na^+^ content in TBT, the stacking of a large number of the rPET molecular chains can lead to entanglement between molecules. This will slightly weaken the mobility of the rPET molecular chains, resulting in a slight decrease in the mechanical properties of the rPET/TBT, but it is basically equivalent to the rPET/DT.

## 3. Materials and Methods

### 3.1. Materials

The rPET (rPET-PCR80AP) was supplied by the Ningbo Jianfeng New Material Co., Ltd. (Ningbo, China). The benzoic acid, the terephthalic acid, and the 1,3,5-benzenetricarboxylic acid were purchased from Shanghai Macklin Biochemical Co., Ltd. (Shanghai, China). The sodium hydroxide (NaOH) was purchased from the Shanghai Aladdin Biochemical Technology Co., Ltd. (Shanghai, China). The acetone, alcohol, phenol, and tetrachloroethane are purchased from Tianjin Tianli Chemical Reagent Co., Ltd. (Tianjin, China).

### 3.2. Preparation of the Sodium Benzoate Salt Nucleating Agent

Benzoic acid (0.164 mL), terephthalic acid (0.012 mol), and 1,3,5-benzenetricarboxylic acid (0.009 mol) were dissolved in 100 mL of ethanol, respectively. Then, NaOH (0.164 mol, 0.006 mol, and 0.003 mol) was dissolved in 30 mL of deionized water, respectively. Subsequently, the NaOH solution was added drop by drop into the ethanol solution with a controlled dropping time of 0.5 h. The mixture was stirred at 65 °C for 6 h to obtain the reaction system. After the reaction was completed, 100 mL of the acetone solution was added to the reaction system to ensure complete precipitation of the crude product. Then, the crude product was washed to neutral with ethanol and deionized water successively, and dried to constant weight at 80 °C to obtain SB, DT, and TBT.

### 3.3. Characterization of the Sodium Benzoate Salt Nucleating Agent

The molecular structures of the sodium benzoate salt nucleating agent and rPET/sodium benzoate salt nucleating agent blend were characterized using an FTIR spectrometer (Nicolet iS10, Thermo Scientific Inc., Waltham, MA, USA), with 64 scans conducted for each sample.

The thermogravimetric (TG) analysis was performed on a TGA (TGA-1, Mettle Toledo Company, Zurich, Switzerland). The stability of the nucleating agent was tested by heating from 40 °C to 600 °C under an N₂ atmosphere at a heating rate of 10 °C/min.

The POM observations of the rPET/sodium benzoate salt nucleating agent were carried out using a polarized optical microscope (DM2700, Leica, Wetzlar, Germany) and combined with a hot stage (THMS 600, Linkam, UK) to control the temperature. The temperature program was set as follows: the rPET/sodium benzoate salt nucleating agent was first heated from 40 °C to 280 °C at a rate of 10 °C/min and maintained at 280 °C for 3 min. Then, it was cooled a rate of 40 °C/min to 225 °C and maintained at 225 °C for 45 min.

### 3.4. Preparation of the rPET/Sodium Benzoate Salt Nucleating Agent Blend

The formula of the rPET/sodium benzoate salt nucleating agent blend was presented in Table 6. The rPET was dried at 80 °C for 12 h and the nucleating agent was dried at 85 °C for 5 h before processing. The rPET/sodium benzoate salt nucleating agent was mixed homogeneously in a high-speed mixer (Songqing Hardware Factory, Yongkang, China, SL−500A). The homogeneous rPET/sodium benzoate salt nucleating agent mixture was put into a twin-screw extruder (Shanghai Xinshuo Precision Machinery Co., Ltd. WLG10A, Shanghai, China) at 280 °C for extrusion and granulation. The tensile, flexural, and Izod notched impact standard test specimens of the rPET/sodium benzoate salt nucleating agent blend were molded using an injection machine (WZS10D, Shanghai Xinshuo Precision Machinery Co., Ltd., Shanghai China) according to GB/T 1040.3-2006 [48], GB/T 9341-2008 [49], and GB/T 1843-2008 [50], respectively. The mold temperature was 40 °C and the cooled time was 1 min. The maximum injection pressure was 2 MPa, and injection molding was carried out at 0.5 MPa for 5 s.

### 3.5. The Viscosity Testing of the rPET/Sodium Benzoate Salt Nucleating Agent Blend

Twenty positions were selected for sampling from each injection-molded specimen. Then, the selected sample points were mixed and weighed to 0.125 g (with an accuracy within ±0.005 g). And a 1:1 (mass ratio) solution of phenol tetrachloroethane should be prepared in a thermostat at (25 ± 0.1) °C. Next, 15 mL of the phenol-tetrachloroethane solution was added to the conical flask using a pipette, and the mixture was heated in an oil bath at 110 °C for 30 min until the sample was completely dissolved. Subsequently, the sample should be cooled to room temperature. The viscosity test was carried out by placing the UST viscometer completely in a water bath at 25 °C. The efflux times of the pure solvent and the sample solution were determined separately. The above procedure was repeated five times, and the average value was taken.
(5)$$\eta_{r}=\frac{t_{1}}{t_{0\;}}$$
(6)$$\eta_{\mathrm{sp}}=\eta_{r}-1$$
(7)$$\left[\eta \right]=\frac{\sqrt{1+1.4\eta_{\mathrm{sp}}}-1}{0.5C}$$
(8)$$\left[\eta \right]=KM_{\eta }^{\alpha }$$
where *t*_0_ is the solvent efflux time, *t*_1_ is the solution efflux time, *η_r_* is the relative viscosity, and *η_sp_* is the incremental viscosity; [*η*] is the intrinsic viscosity, and *C* is the solubility of the specimen in solution. $M_{\eta }$ is the Visco-average molecular weight. *K* and *α* are the Mark–Houwink constants, with values of 2.1 × 10^−4^ dL/g and 8.2, respectively [51].

### 3.6. Characterization of the rPET/Sodium Benzoate Salt Nucleating Agent Blend

The torque test was carried out and the torque with time was recorded using a torque rheometer (XSS-300, Shanghai Kechuang Rubber Machinery Equipment Co., Ltd., Shanghai, China) according to the instructions provided by the manufacturer. The reaction temperature is 270 °C, the rotor speed was 60 r/min, and the test time was 5 min.

The crystallization behavior of rPET/sodium benzoate salt nucleating agent was tested by DSC (Q1000, Waters Corporation, Milford, CT, USA) under nitrogen atmosphere (50 mL/min) protection. The test samples were first heated from 40 °C to 280 °C at a rate of 10 °C/min and maintained at 280 °C for three minutes to eliminate thermal history. Secondly, they were cooled from 280 °C to 40 °C at a rate of 10 °C/min. Next, they were heated up again from 40 °C to 280 °C at a rate of 10 °C/min. Afterward, they were cooled again at a rate of 40 °C/min from 280 °C to 225 °C and maintained at isothermal temperature for 90 min. Ultimately, they were cooled from 280 °C to 40 °C at a rate of 10 °C/min. The heating and cooling curves of each stage were recorded. 

The tensile strength of the rPET/sodium benzoate salt nucleating agent blend was assessed following the GB/T 1040.3-2006, using a universal testing machine (TY−8000A, Jiangsu Tianyuan Test Instrument Co., Ltd., Nanjing, China) at a speed of 20 mm/min. The flexural strength of the rPET/sodium benzoate salt nucleating agent blend was performed according to GB/T 9341-2008, using the universal testing machine at a speed of 5 mm/min. The Izod notched impact strength of the rPET/sodium benzoate salt nucleating agent blend was assessed according to GB/T 1043.1-2008 [52], using a universal testing machine with an impact energy of 5.5 J, an impact angle of 150°. We recorded the results of eight tests on each sample and took the average.

## 4. Conclusions

In this work, three kinds of the sodium benzoate salt nucleating agents (SB, DT, and TBT), which possess the same organic carboxylic acid structure but different Na⁺ contents, were synthesized through a chemical reaction, and the rPET/sodium benzoate salt nucleating agent blend was prepared by melt blending.

The TG results of the sodium benzoate salt nucleating agent proved that the chemical structure of the nucleating agent can remain intact at the processing temperature of rPET (280 °C). The FTIR results were confirmed the successful synthesis of the nucleating agent. Meanwhile, characteristic absorption peaks of the -COO^−^ were observed at 1552 cm^−1^ and 1460 cm^−1^, which indicated that chemical reaction occurred between the sodium benzoate salt nucleating agent and rPET. The results provided condition for the occurrence of chemical nucleation, which offered the possibility of improving the crystallization properties of rPET.

Moreover, the intrinsic viscosity data of the rPET/sodium benzoate salt nucleating agent blend indicated that the *η* of the rPET/SB decreased by 0.12 dL/g compared to the rPET. This result suggested that chemical reaction between the Na^+^ of SB and the rPET led to the breakage of the rPET molecular chains and a decrease in molecular weight. However, the *η* of the rPET/DT and rPET/TBT increased by 0.15 dL/g and 0.33 dL/g, respectively. The reason for this phenomenon may be that the Na^+^ contents of the DT and the TBT were higher than that of the SB, increasing the chance for the low-molecular-weight rPET to reattach to the organic carboxylic acid portion of the DT and TBT. Meanwhile, the torque results of the sample can also prove this conclusion. This also provided the possibility for improving the mechanical properties of rPET. Compared with the rPET, the *T_cc_* of the rPET/SB, rPET/DT, and rPET/TBT decreased by 8.89 °C, 3.86 °C, and 1.83 °C, respectively. The *T_c_* increased by 26.03 °C, 14.6 °C, and 9.85 °C, respectively. The *t*_1/2_ shorten by 12.11 min, 8.54 min, and 4.71 min, respectively. This result indicated that chemical nucleation had appeared in the rPET/sodium benzoate salt nucleating agent blend. The ionic group generated by chemical nucleation serves as the nucleation site, which can induce more crystal nucleation and greatly improve the crystallization properties of rPET.

Furthermore, the chemical nucleation between SB and rPET caused the local degradation of the rPET molecular chains. This led to stronger mobility, thereby improving the crystallization properties and the crystallization rate of rPET. Although the rPET/DT and rPET/TBT had a high molecular weight, they also limited the crystallization rate of the blend to some extent. Therefore, the improvement of the rPET crystallization behavior by DT and TBT was slightly worse than the SB. In addition, the Avrami index *n* demonstrated that the addition of nucleating agent had no effect on the crystal growth mode of rPET. The POM result indicated that the addition of the nucleating agent effectively improved the uniformity of rPET grains, increasing the number and density of spherulites. A denser and more uniform crystal structure would weaken the molecular chains mobility of the rPET, thereby enhancing its ability to disperse tensile stress under external forces, which might also improve the mechanical properties of the blend.

The mechanical properties of the rPET/sodium benzoate salt nucleating agent blend indicated that the tensile strength of the rPET/SB decreased by 12.26 MPa, the flexural strength decreased by 25.96 MPa, the impact strength decreased by 0.22 KJ/m^2^, and the flexural modulus increased by 406.93 MPa compared to the rPET. This was because chemical nucleation caused the partial degradation of the rPET, resulting in a decrease in mechanical properties. Compared to rPET, the tensile strength of the rPET/DT and rPET/TBT with a higher relative molecular weight increased by 1.14 MPa and 0.94 MPa, the flexural strength increased by 4.27 MPa and 2.37 MPa, and the flexural modulus increased by 128.9 MPa and 109.28 MPa, respectively. However, when a large number of the rPET molecular chains were stacked near the ion cluster, the rPET/TBT with the highest molecular weight will form a structure similar to physical cross-linking. This would hinder the movement of the rPET molecular chains, weaken their ability to enter the lattice, and result in a slight decrease in mechanical properties compared to the rPET/DT.

From an engineering perspective, the addition of the nucleating agent increased the crystallization temperature of rPET and shortened its molding cycle, thereby improving production efficiency. By optimizing the crystal structure of rPET, it was endowed with more excellent mechanical properties, enabling the manufacture of higher-quality products and thus expanding its application range in the field of engineering plastics. This further promoting the feasibility of PET recycling and reuse, aligning with the development concept of green ecology.

## Figures and Tables

**Figure 1 molecules-30-00037-f001:**
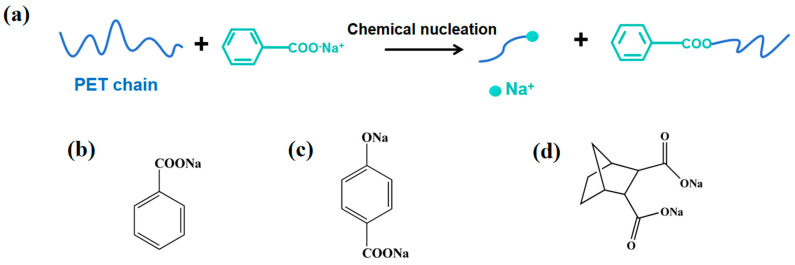
(**a**) The nucleation mechanism of the PET reaction with the Na^+^; (**b**) the molecular structure of the SB; (**c**) the molecular structure of the PHB-2Na; and (**d**) the molecular structure of the HPN-68L.

**Figure 2 molecules-30-00037-f002:**
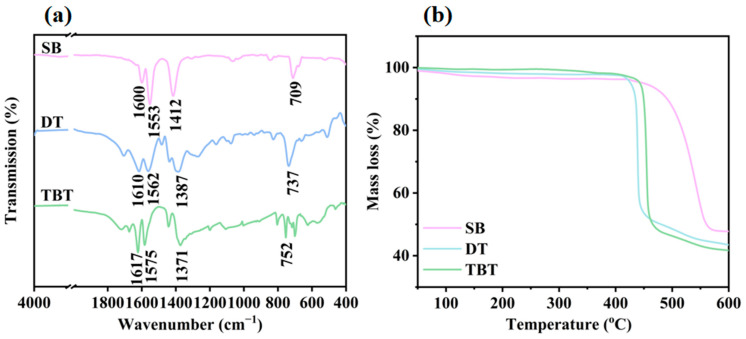
(**a**) The FTIR spectra of the sodium benzoate salt nucleating agent; and (**b**) the TG spectra of the sodium benzoate salt nucleating agent.

**Figure 3 molecules-30-00037-f003:**
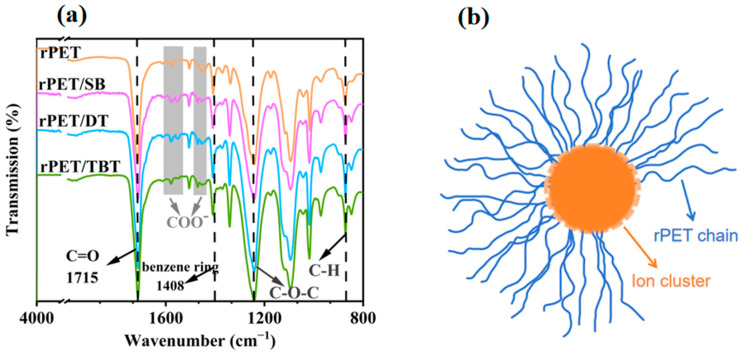
(**a**) The FTIR spectra of the rPET/sodium benzoate salt nucleating agent blend; and (**b**) the electrostatic interaction between COO^−^ and Na^+^ at the end of rPET molecular chains forms ion clusters.

**Figure 4 molecules-30-00037-f004:**
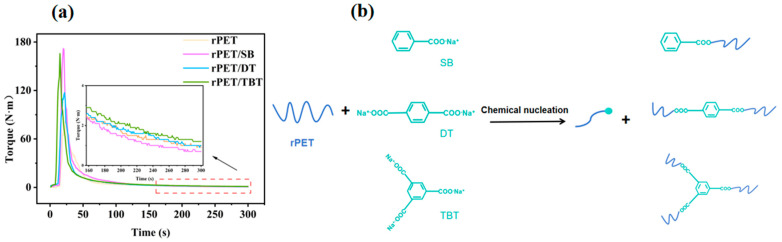
(**a**) The torques of the rPET/sodium benzoate salt nucleating agent blend; and (**b**) the reaction mechanism of the sodium benzoate salt nucleating agent with the rPET.

**Figure 5 molecules-30-00037-f005:**
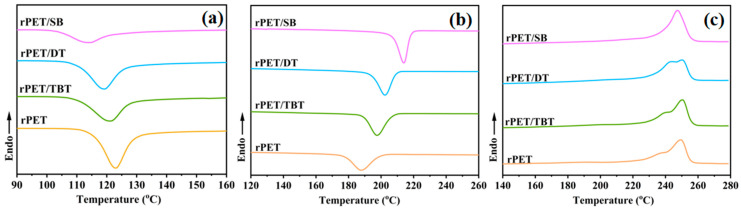
(**a**) The first melting curve of the rPET/sodium benzoate salt nucleating agent blend; (**b**) the non-isothermal crystallization curve of the rPET/sodium benzoate salt nucleating agent blend; and (**c**) the second melting of the rPET/sodium benzoate salt nucleating agent blend.

**Figure 6 molecules-30-00037-f006:**
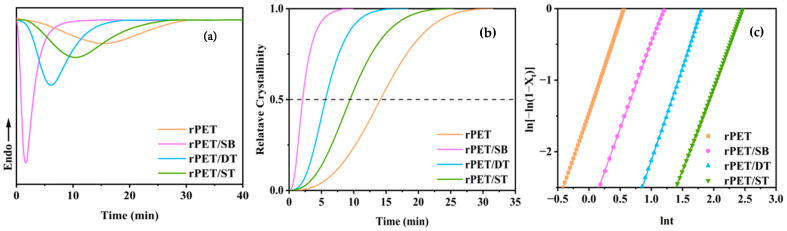
(**a**) The isothermal crystallization curve of the rPET/sodium benzoate salt nucleating agent blend; (**b**) relative degree of crystallinity with time of the rPET/sodium benzoate salt nucleating agent blend; and (**c**) Avrami curves of the rPET/sodium benzoate salt nucleating agent blend.

**Figure 7 molecules-30-00037-f007:**
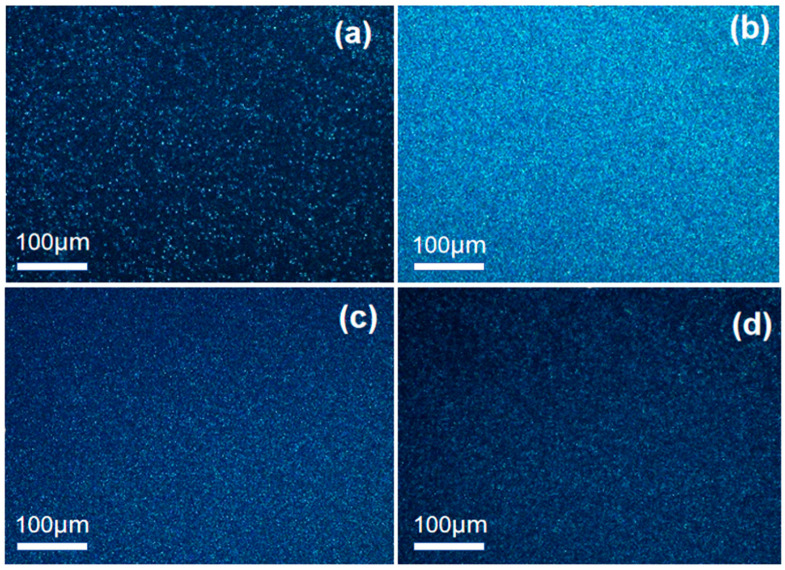
The POM morphology of samples at 225 °C: (**a**) rPET; (**b**) rPET/SB; (**c**) rPET/DT; and (**d**) rPET/TBT.

**Table 1 molecules-30-00037-t001:** The dissociation constant and solubility parameter of the acid corresponding to the sample.

Sample	δ/Solubility Parameter (cal/cm^3^)^1/2^
rPET	10.6
benzoic acid	11.0
terephthalic acid	11.9
trimesic acid	12.3

**Table 2 molecules-30-00037-t002:** The intrinsic viscosity of the rPET/sodium benzoate salt nucleating agent blend.

Sample	Time(s)					[*η*]/(dL/g)	*M_η_* (g/mol·10^4^)
phenol/tetrachloroethane	90.1	89.9	90.0	90.1	89.9		
rPET	115.6	115.8	115.6	115.7	115.6	0.522 ± 0.003	1.38 ± 0.1
rPET/SB	109.1	109.2	109.0	109.1	109.1	0.40 ± 0.003	0.99 ± 0.1
rPET/DT	123.5	123.7	123.6	123.7	123.7	0.67 ± 0.003	1.87 ± 0.1
rPET/TBT	133.9	133.7	133.7	133.8	133.9	0.85 ± 0.002	2.49 ± 0.1

**Table 3 molecules-30-00037-t003:** Non-isothermal crystallization parameters of the rPET/sodium benzoate salt nucleating agent blend.

Sample	*T_cc_* (°C)	*T_c_* (°C)	*T_m_* (°C)	*X_c_ (%)*
rPET	122.92	187.64	249.54	28.2
rPET/SB	114.03	213.67	247.89	33.7
rPET/DT	119.06	202.24	250.40	31.7
rPET/TBT	121.09	197.49	250.67	30.7

**Table 4 molecules-30-00037-t004:** The isothermal crystallization kinetic parameters of the rPET/sodium benzoate salt nucleating agent blend.

Sample	*n*	*K* (min^−n^)	*t*_1/2_ (min)
rPET	2.58	6.6 × 10^−4^	14.18
rPET/SB	2.41	1.3 × 10^−1^	2.07
rPET/DT	2.59	3.3 × 10^−3^	5.64
rPET/TBT	2.31	7.2 × 10^−3^	9.47

**Table 5 molecules-30-00037-t005:** The mechanical properties of the rPET/sodium benzoate salt nucleating agent blend.

Sample	Tensile Strength (MPa)	Flexural Strength (MPa)	Flexural Modulus (MPa)	Izod Notched Impact Strength (kJ/m^2^)
rPET	50.18 ± 1.19	78.93 ± 1.19	2260.35 ± 43.11	1.26 ± 0.04
rPET/SB	37.92 ± 1.14	52.97 ± 1.04	2067.28 ± 24.93	1.03 ± 0.04
rPET/DT	51.32 ± 1.27	83.20 ± 1.27	2389.25 ± 40.29	1.37 ± 0.05
rPET/TBT	51.12 ± 1.23	81.30 ± 1.03	2382.63 ± 35.52	1.31 ± 0.04

**Table 6 molecules-30-00037-t006:** The formula of the rPET/sodium benzoate salt nucleating agent blend.

Sample	rPET (wt%)	SB (wt%)	DT (wt%)	TBT (wt%)
rPET	100	0	0	0
rPET/SB	100	0.5		
rPET/DT	100		0.5	
rPET/TBT	100			0.5

## Data Availability

The data are contained within the article.

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
