# Peer review of "The Application of a Sodium Benzoate Salt-Nucleating Agent in Recycled Polyethylene Terephthalate: Crystallization Behavior and Mechanism"

_molecules, 2024, doi:10.3390/molecules30010037_

Round 1

Reviewer 1 Report

Comments and Suggestions for Authors

The manuscript titled “The Application of the Sodium Benzoate Salt Nucleating Agent in Recycled Polyethylene Terephthalate: Crystallization Behavior and Mechanism” addresses an important and timely topic. However, improvements are required in several areas.

Structure:

The article's structure needs to be adjusted: the Materials and Methods section should be placed before the Results section.

Introduction:

The statement, “Most importantly, thermal degradation and photolysis during the daily use of PET can lead to a decrease in the molecular weight of PET, which results in a decline in the mechanical and crystallization properties of the rPET [13].”

is controversial in the literature. It should also include a discussion of how many researchers argue that shorter molecular chains promote rearrangement, leading to a higher crystalline fraction in recycled PET compared to the original material.

Given that one of the nucleating agents demonstrates a chain-extending effect in the results — which is a very significant finding! — the Introduction should address chain extenders for PET and their effects (e.g., on melt strength).

Materials and Methods:

The preparation of test specimens needs to be described in more detail.

•             Was drying applied before each processing step? If yes, what were the parameters?

•             What does "Standard test specimen" mean? Which standard does it follow, and what was the geometry?

•             To what does the 0.5 MPa refer in injection molding (maximum injection pressure/holding pressure)?

•             Importantly, what was the mold temperature, and how long was the cooling time? These factors heavily influence the crystallinity of the produced test specimens. For instance, according to Table 3, the crystallinity of rPET without nucleating agents is already quite high (28%), which suggests the use of a warm mold.

Are the intrinsic viscosity (IV) measurement results reliable? What was the repeatability/SD of these measurements?

Do the values in Table 2 come from the injection-molded samples? What was the starting IV value of the base rPET material? Was the base material sourced from ground bottles or some other waste source?

The description of mechanical tests is entirely missing! What standards, instruments, and methods were used for these tests? Were impact tests conducted using the Charpy or Izod method, and were the samples notched or unnotched?

The recorded impact strength values are consistently very low (<1.4 kJ/m²). To what extent might this affect the usability of nucleating agents?

Results:

•             Figure 7 does not clearly illustrate the intended phenomenon. Higher magnification images with better contrast should be provided.

•             Figure 8 represents the measurement results as a function, but the x-axis is not meaningful. This should be replaced with a bar chart since the results are not presented as a function of a variable.

•             Figure 8 and Table 5 are duplicates; it is sufficient to present one of them.

The evaluation of modulus results is missing in the mechanical tests. This is essential, as crystallinity often influences this property. The evaluation must be added.

How did the nucleating agents affect the optical properties/aesthetics of the specimens?

Minor Errors:

1.            A space should be added before all citations, e.g., [13].

2.            Table 3 title: Correct "nucleating" from "nu cleating".

3.            In Table 5 (4th column) and in several places in the text, "kJ/m²" should replace "KJ/m²" (the "kilo" prefix should use a lowercase "k").

Conclusion:

From an engineering perspective, what was achieved with the application of nucleating agents? How do the additives influence the usability of recycled PET in different fields? Have they opened any new possibilities that support the goals of sustainable development?

Reviewer 2 Report

Comments and Suggestions for Authors

Dear authors 

The paper explores the crystallization behavior of rPET when modified with various sodium benzoate salt nucleating agents. The results are supported by substantial experimental evidence, including FTIR, TGA, DSC and mechanical tests. There are clear efforts to link the experimental findings to theoretical mechanisms. The paper is technically sound and presents valuable insights into the crystallization behaviour of rPET. I suggest a few improvements that would significantly increase its impact and clarity:  

  • Some sentences lack grammatical coherence, e.g., the use of phrases like "chemical reaction was occur" instead of "chemical reaction occurred." 

  • The paper contains frequent grammatical errors (e.g., "was indicated" instead of "indicates"). 

  • Style inconsistencies occur, such as switching between past and present tense in the same paragraph. 

  • The "Materials and Methods" section, while detailed, contains unnecessary repetitions that could be streamlined. 

  • Paragraphs sometimes blend experimental results with interpretations without clear transitions, reducing readability. 

  • The discussion section could more explicitly compare findings with other studies to emphasize the novelty of the results. 

  • The mechanical property analysis could delve deeper into why certain nucleating agents lead to decreased strength despite improved crystallinity. 

  • Some claims, such as the improvement of rPET crystallization behavior, would benefit from additional comparison to prior research. 

  • The conclusion is concise but misses an explicit discussion of the study's broader implications or potential applications. 

  • Future research directions are briefly mentioned but not well-articulated. 

  • The discussion on environmental implications of improved rPET recycling could be expanded. 

Reviewer 3 Report

Comments and Suggestions for Authors

The textual description should be further optimized, as there are many typographical and grammatical errors. The image content and layout need further improvement. The explanation of some conclusions is not clear enough. Please refer to the following Word document for specific suggestions.

Comments on the Quality of English Language

There are many grammar and writing errors in English, which should be further corrected.

Round 2

Reviewer 1 Report

Comments and Suggestions for Authors

The corrections to the manuscript are correct, except for one small detail: When specifying the test specimens used, the standards applied are still not given. This should be completed. If the standards are given, it is not necessary to show the specimen (Fig. 8) as the standards clearly control the geometry.

Author Response

1. The corrections to the manuscript are correct, except for one small detail: When specifying the test specimens used, the standards applied are still not given. This should be completed. If the standards are given, it is not necessary to show the specimen (Fig. 8) as the standards clearly control the geometry.

reply: Thank you for your arduous work and instructive advice. According to the reviewer's suggestion, I have added the standards for the sample and deleted Figure 8. All revisions have been added to the manuscript and highlighted in red font.

Reviewer 3 Report

Comments and Suggestions for Authors

I read the revised manuscript carefully and believe there are no problems.

Author Response

We are very grateful for your helpful suggestions for revision. Thank you for your approve of my article.